# 'I wanted to go, but they said wait': Mothers' bargaining power and strategies in care-seeking for ill newborns in Ethiopia

**Kristine Husøy Onarheim**[1,2,3]*, **Karen Marie Moland**[1,3], **Mitike Molla**[4], **Ingrid Miljeteig**[1,5]

**1** Department of Global Public Health and Primary Care, University of Bergen, Bergen, Norway, **2** Institute for Global Health, University College London, London, United Kingdom, **3** Centre for Intervention Science in Maternal and Child Health, University of Bergen, Bergen, Norway, **4** School of Public Health, College of Health Sciences, Addis Ababa University, Addis Ababa, Ethiopia, **5** Department of Research and Development, Helse Bergen Health Trust, Bergen, Norway

* kristine.onarheim@uib.no

**Data Availability Statement:** The study is based on qualitative data from interviews and FGDs. The interview transcripts cannot be made publicly

## Abstract

### Introduction

To prevent the 2.6 million newborn deaths occurring worldwide every year, health system improvements and changes in care-taker behaviour are necessary. Mothers are commonly assumed to be of particular importance in care-seeking for ill babies; however, few studies have investigated their participation in these processes. This study explores mothers' roles in decision making and strategies in care-seeking for newborns falling ill in Ethiopia.

### Methods

A qualitative study was conducted in Butajira, Ethiopia. Data were collected during the autumn of 2015 and comprised 41 interviews and seven focus group discussions. Participants included primary care-takers who had experienced recent newborn illness or death, health care workers and community members. Data were analysed using thematic analysis.

### Results

Choices about whether, where and how to seek care for ill newborns were made through cooperation and negotiation among household members. Mothers were considered the ones that initially identified or recognised illness, but their actual opportunities to seek care were bounded by structural and cultural constraints. Mothers' limited bargaining power, contained by financial resources and gendered decision making, shaped their roles in care-seeking. We identified three strategies mothers took on in decision making for newborn illness: (a) acceptance and adaptation (to the lack of options), (b) negotiation and avoidance of advice from others, and (c) active care-seeking and opposition against the husband's or community's advice.

### Conclusion

While the literature on newborn health and parenting emphasizes the key role of mothers in care-seeking, their actual opportunities to seek care are shaped by factors commonly

available as this would compromise ethical concerns for participants' privacy and confidentiality.

**Funding:** This study was funded by the University of Bergen, Norway (personal PhD grant). The funder had no role in study design, data collection and analysis, decision to publish, or preparation of the manuscript.

**Competing interests:** The authors have declared that no competing interests exist.

beyond their control. Efforts to promote care-seeking for ill children should recognise that mothers' capabilities to make decisions are embedded in gendered social processes and financial power structures. Thus, policies should not only target individual mothers, but the wider decision making group, including the head of households and extended family.

## Introduction

Evidence-informed policies to improve newborn health promote early detection and rapid care-seeking for ill babies. The most recent World Health Organization (WHO) guidelines on newborn health (2017) describe when and what health practices a caregiver should adhere to, abandon or avoid [1]. Research demonstrates the intimate links between maternal and child health before, during and after pregnancy [2]. The survival of the mother following birth is crucial for the survival of the infant [3]. With the aim of averting the 2.6 million newborn deaths that occur every year [4–6], policies target primary care-takers. The important role of the mother in health care-seeking for ill newborns is embraced in both international and national health policies and guidelines [5–8]. However, as we will argue in this paper, the responsibility placed on the mother as the key actor to improve newborn health and in care-seeking for ill babies may not be supported by her capability to fill this function. Health care-seeking for ill newborns is influenced by determinants beyond the mother's control, such as advice and practices of community members, which in turn are shaped by cultural norms and socioeconomic factors [9–11]. Further, newborn health policies may encounter challenges in their adoption and implementation in diverse communities, social settings and health systems [12].

Intra-household decision making is relevant for our understanding of the role of mothers in care-seeking for ill babies. Recent research provides nuance to how mothers take part in these decisions and how gendered household roles restrict their options [9,10,13]. A study from Mali revealed that men and women played different roles in treatment decisions for ill children with malaria [14]. The main decision making responsibilities shifted to fathers and senior household members when the child's condition was perceived as severe. While illness identification was primarily considered as the responsibility of mothers (52%) and senior women (24%), fathers were responsible for payment (75%). Less than 8% of informants thought of mothers as the most capable in health care decision making outside the household [14]. In Ghana, women who disagreed with other family members or with limited economic means experienced difficulties in accessing health care for their ill children [15]. Related to these findings, a narrative review examining intra-household bargaining power and processes highlighted how gender influence child health and nutrition through (a) women's decision making power and access to and control over resources and (b) household headship [9]. This literature illustrates how different positions and power within and beyond the household influence decisions about whether and how to seek health care [9,15,16]. As policies to improve newborn health target mothers, it is critical to examine their bargaining power–the relative power of parties in a situation to exert influence over each other–in care-seeking. Further exploration of how decisions are made in particular contexts is necessary if we are to understand why some seek health care services and others do not. While studies emphasize the importance of household and family-level factors and early recognition of child illness [17–19], little is known about the roles and strategies taken by the mother in real-life decision making. We argue that this requires specific attention, as it is often assumed—explicitly or

implicitly—that the mother as the primary care-taker will take action once she has identified danger signs of newborn illness [6,7].

## Newborn health in Ethiopia

This study was conducted in Ethiopia, where highly gendered divisions of labour and power dynamics at household level shape everyday life and decision making processes [20]. In 2018, Ethiopia was ranked as number 117 of 149 countries on the Global Gender Gap Index 2018 [21]. Many women lack economic autonomy and employment rates differ starkly, where 48% of women and 99% of men had been employed the previous year. At home, 71% of women are actively involved in household decision making on health care, household purchases and visits to their family [22]. These gendered roles in private and public life are likely to influence mothers' care-seeking.

Ethiopian health policy reforms have given priority to maternal and child health and to scaling-up services and providing a continuum of care [7,8]. Still, mortality is falling slower for newborns than for older children and the number of newborn deaths remains high (61,600 deaths in 2017) [4,23]. The Ethiopian newborn and child survival strategy (2015–2020) presents key steps to reduce neonatal mortality from 28 to 11 deaths per 1000 live births (2013–2020) [7], with emphasis on high impact intervention packages and increasing coverage, quality and equity [8]. One of the components to improve newborn health is community-based newborn care. Community-based newborn care aims to identify and treat illness through community mobilization and is delivered through the Health Extension Programme (primary health care programme) and the Women's Development Army (community-based programme). Despite political priority, coverage of newborn and maternal health services is low and out-of-pocket expenditures are high [22,24]. An Ethiopian study found that 81% of newborn deaths were associated with delays in treatment seeking outside the home [25]. Hence, attention to demand-side factors outside health facilities, including mothers' roles in decision making and care-seeking for newborns, seems warranted. In this article we explore the roles of mothers in decision making for newborns falling ill in Ethiopia and the strategies they employ in care-seeking processes.

## Methods

A larger qualitative study on decision making and resource allocation on newborn health was conducted in Ethiopia in the autumn of 2015 [26,27]. The study was carried out in Butajira, located 130 km south of the capital Addis Ababa. Skilled delivery and postnatal check-ups in the region is low (26% and 17%, respectively), but similar to national coverage [22,28]. Four out of ten women in Butajira have experienced child death [29]. Studies from this poverty-stricken context has shown delays in care-seeking for ill newborns, when they were perceived to be at high risk of dying and health care-seeking was costly [26,27].

The study was carried out through the Butajira Rural Health Program (BRHP). The BHRP is a health and demographic surveillance system which has monitored and collected data on demographics and vital events in its ten *kebeles* (villages) in the area since 1987 [30]. Participants were recruited from the BHRP catchment area and two hospitals and affiliated health centers surrounding Butajira. The data comprised 41 in-depth interviews with primary care-takers and health care workers and seven focus group discussions (FGDs) with community members and health care workers (Table 1). Triangulation in sources of participants (primary care-takers, health care workers, community members) and data collection methods (interviews, FGDs) and member checking during data collection were conducted to enhance the study's trustworthiness and credibility.

With emphasis on decision making processes and care-seeking, in-depth interviews (11) in hospital with primary care-takers experiencing newborn illness (admitted at hospital > one day) were followed by subsequent interviews at home (9). Through retrospective identification of cases of newborn deaths the previous year, we talked to primary care-takers (10) about their experiences when they baby had fallen ill. Baseline characteristics of primary care-takers are described in Table 2. To better understand the collective aspects of decision making processes and care-seeking, we approached community members (5 FGDs) and health care workers (11 interviews and 2 FGDs) (Table 1). Identification of community members was done in collaboration with BHRP independent of recruitment of primary care-takers.

Interviews and FGDs were audio-recorded (all FGDs, 38 interviews) or notes were taken (three interviews). Transcripts were translated from Amharic to English. One of the authors conducted quality checks in translation. Data were analysed using thematic analysis [31]. Initial analysis started during data collection with daily reviews of topics covered in interviews and FGDs. Interview and topic guides were revised to integrate new insights. The transcribed data material was analysed and coded (NVivo 11 software) to identify core themes and patterns (Fig 1). Overarching themes and sub-themes were discussed within the team after data collection and initial coding of the material. Following Braun and Clark [31, page 87], the analysis involved six phases: I: Getting familiar with the data (all authors). II: Initial coding (led by one author, supported by other author). III: Searching for themes (all authors). IV: Reviewing themes (three authors). V: Defining and naming themes (three authors) (Fig 1). VI: Producing the report (all authors). An inductive-deductive approach was taken in the analysis, going back and forth between transcriptions and the coded data material in phases III-VI [31]. After the descriptive phase of the analysis, where we focused on the manifest and descriptive meanings of the data, we moved on to the interpretative phase where we searched for latent meanings [31, page 84]. For instance, a mother's description of how "*[the neighbours] insisted that he [the baby] should stay at home*" was coded under the sub-theme "depending on guidance and support from neighbours and family members" under the theme "depending on others" (Fig 1). Going beyond manifest descriptions we took an interpretive approach in examining latent features and meanings of mothers' care-seeking decisions. We understood mothers' practices, as when avoiding advice from the elderly, as a manoeuvring strategy in navigating care-seeking. Uncertainty in analysis and interpretation of the data was discussed within the team.

**Table 1. Participants of in-depth interviews and focus group discussions (FGDs).**

| |
|---|
| **Primary care-takers** |
| 11 interviews with primary care-takers of ill newborn at hospital [9 follow-up interviews] |
| 10 interviews with primary care-takers facing newborn death |
| **Health care workers** |
| 3 interviews with medical doctors involved in newborn care |
| 7 interviews with nurses and midwives involved in newborn care |
| 1 interview with informant from health bureau (male) |
| 1 FGD with nurses and midwifes (urban, 6 participants) |
| 1 FGD with health extension workers (rural and urban, 6 participants) |
| **Community members** |
| 1 FGD with women in reproductive age with child <1 year (urban, 8 participants) |
| 1 FGD with women in reproductive age with child <1 year (rural, 4 participants) |
| 1 FGD with husbands with wife in reproductive age with child <1 year (rural, 6 participants) |
| 1 FGD with grandmothers (rural, 6 participants) |
| 1 FGD with religious leaders and elders (urban, 6 participants) |

**Table 2. Basic characteristics of primary care-takers experiencing newborn illness and death.**

| | Primary care-takers experiencing newborn illness | Primary care-takers experiencing newborn death |
|---|---|---|
| **Location** | | |
| Rural | 6 | 5 |
| Urban | 5 | 5 |
| **Working** | | |
| Housewife | 6 | 6 |
| Small business or employed | 5 | 4 |
| **Rooms in house**\* | | |
| 1 | 5 | 3 |
| 2 | 4 | 3 |
| >3 | 1 | 3 |
| **Number of pregnancies**\* | | |
| 1–2 | 7 | 2 |
| 3–6 | 3 | 4 |
| >7 | 0 | 2 |
| **Birth in facility** | | |
| In hospital/health center | 7 | 5 |
| Not hospital/health center | 4 | 5 |
| **Total** | 11 | 10 |

For newborn illness, seven interviews included the mother as the primary care-taker and four interviews included the mother and father as primary care-takers. For newborn deaths, nine interviews included the mother as primary care-taker and one interview included the mother and father as primary care-takers.

\* Not all study participants revealed this information.

Building on the data material and interpretation, final themes and sub-themes were developed (Fig 1).

The study was approved by the Institutional Review Board of the College of Health Sciences, Addis Ababa University (025/15/SPH) and the Regional Ethical Committee of Western Norway, Norway (2015/327/REK vest). Potential participants were informed verbally about the study, the opportunity to participate, that participation was voluntary, and that no personal data would be revealed. Informed consent (written or by fingerprint) was obtained from potential participants before commencing any interviews or FGDs.

The methods are also described elsewhere [26,27].

## Results

In Butajira, we found three central decision points in health care-seeking for ill newborns. First, is there a problem with the baby? Second (if yes), should the baby be taken to a health facility? Third (if yes), how and where should care be sought for the baby? The following sections describe how mothers were involved in these decisions.

### Mothers identify the problem and inform others

Though mothers themselves, health care workers and community members considered mothers crucial in care-seeking for ill newborns, their roles and actual opportunities to make decisions were restricted. Mothers described how they were responsible for identifying illness of the baby and the need for care. They recognised signs of illness, such as "she couldn't suck breast" or when the baby was found too tired, too hot or cold, weak, screamed "unnatural" or had abnormal breathing. Some had noticed that the baby was unwell at the time of birth;

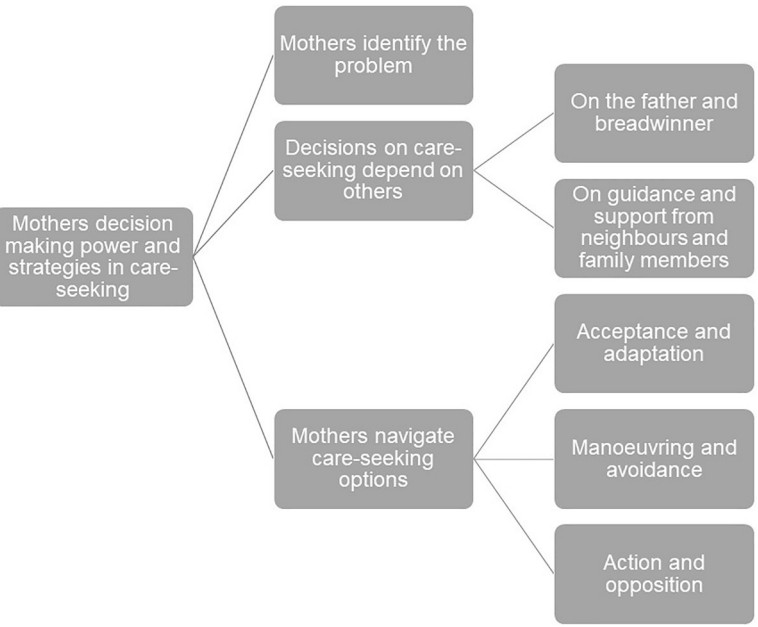

**Fig 1. Themes and sub-themes.**

others reported that the baby's condition altered during the first days following birth. As one mother explained, she was alerted when her baby stopped sucking:

> He was breastfeeding at that time [after birth], but he suddenly stopped. He got really sick. I told them [the neighbours]. (Interview, mother experiencing newborn death, rural, 17)

In Butajira and surrounding areas, women are often cared for in their natal family's home around the time of birth, and our interviews revealed that husbands rarely were present following birth. Mothers, sisters, other relatives and elderly in the close surroundings were often involved in care for the mother and baby following birth. They also provided advice or instructions on the health and well-being of the baby, including whether and how to seek health care. Still, in interviews with mothers, health care workers and FGDs with community members the importance of the mother in care-seeking processes was reiterated, particularly in informing the father if a baby was unwell. If the mother had access to a mobile phone, she would call him, or relatives used their networks to find him. At times, this could take many hours and sometimes their efforts failed. Reaching the husband was necessary to inform him of the baby's condition and for him to decide whether and where to seek care, a decision often hinging on costs. Hence, mothers' observations and reporting were key in the defining that the baby was ill (first decision point).

## Depending on others

While mothers commonly experienced babies falling ill, decisions about whether, where and how to seek care were not necessarily theirs to make. In the hospital and follow-up interviews, mothers explained that it was the husband and wife together or the husband alone who had the final say about whether to go to the hospital or health center.

> My husband make money as a daily labourer. He was afraid that we might lose the child, so we brought her here. It was his decision [to seek health care at the hospital], not mine. . . I

also wanted to get involved; I am poor and not [financially] capable by myself, apart from the small income I get from selling [food] at the market. (Interview, mother experiencing newborn illness, rural, 4)

FGD participants described the importance of the father's decision making role and power.

Since the man is the head of the household the decision is expected from him. The woman should discuss about the child illness so that the man could see that the child is really ill. The mother is the one who is close to the child—more so than the father. She should immediately explain about the illness and the father will decide to take the child to the health facility. (FGD, husbands, rural)

Health care workers explained that it was the breadwinner–most often the husband–who ultimately decided whether or not to bring the baby to the hospital and which care that was preferred: "It's the husband's role because he is the income generator of the house." Health care workers repeatedly described how the father and husband's final judgement was most important:

Most often it is the men who decide whether to go home or to stay [at the hospital when the baby is ill]. They are given the highest importance in the family and because of that, they decide on everything. The women also obey their husbands' decisions and instructions. . . Say a mother wants to take her baby to the hospital, but the husband refuses to take them. Since he's the decision maker, he can even take the baby home. (Interview, health care worker, urban, 23)

Beyond the household, neighbours and relatives played vital roles as direct and indirect advisors. They also assisted by lending money or organising transportation. Based on earlier experience, older relatives or neighbours often thought they knew whether and where to seek treatment. One mother explained how the neighbours advised her when she gave birth to twins alone. When one of the babies was found ill, the neighbours suggested he was too sick to be taken to the hospital.

Some of them [neighbours] wanted to take him to the hospital. But most of them refused and insisted that he should stay at home. . . They also advised me not to take him to the health care facility. They said it would be hard to inject medicine on the infant because he is small. (Interview, mother experiencing newborn death, rural, 17)

In this poverty-stricken context, mothers were commonly recommended to "wait and see" if the baby's health improved and care-seeking was often delayed. Mothers, health care workers and community members explained how poverty and social expectations shaped care-seeking and that the costs of medical care could be high and risky for the whole family [26,27]. They had to pay out-of-pocket to cover the medicines, diagnostics and hospital stay and worried about how these expenses would affect the rest of the family. The mother was most often economically dependent on her husband or others to seek care, and women were not expected to make decisions on care-seeking themselves.

Local knowledge, practices and expectations following child birth also influenced care-seeking, both for the baby and the mother. During the first weeks and months after birth the mother should stay at home, a period known as *aras be*t. Religious leaders and women living in rural areas explained how the mother is not expected to leave the house, even if the child

becomes ill (FGD, women rural, FGD, religious leaders*). These norms and expectations made care-seeking at the hospital less likely. As one follow-up interview revealed:

> Taking the neonate to the hospital might expose the mother to sunlight. It is not good for the mother to leave the house before you complete 15 days, it is better to stay for one month. Since there is such kind of belief, people often use traditional medicines rather than taking [the baby] to hospital. . . Here, it's not considered good to take out neonates before christening. It is very scary before 40 days. (Follow-up interview, newborn illness, rural, 32)

Hence, decisions about whether to seek care (second decision point) and how and where to seek care (third decision point), were not made by the mother alone. The outcomes of these decisions affected the whole family and the extent of the mother's involvement in decision making often hinged on intra-household power dynamics alongside economic and cultural factors.

## Navigating care-seeking options

Although decisions about care-seeking depended on the head of household and the main role of the mother was to recognise and report illness, our analysis showed that within their rather narrow space of action, mothers found ways to navigate care-seeking options. We identified three main strategies that mothers adopted: (a) acceptance and adaptation, (b) negotiation and avoidance and (c) action and opposition.

**Acceptance and adaptation.** When a baby fell sick, mothers were expected to listen to their husbands, older relatives or the elderly. Poor mothers residing in areas far from health centers and hospitals had little choice besides adapting to the situation, which often meant staying at home. In rural settings, mothers commonly waited to see if the baby's health improved. While mothers expressed that they had hoped that the baby would get better, others explained how they had seen that the baby was "weak." Often sick or tired herself after the labour and with health facilities hard to reach, mothers did not consider the health center or hospital as viable options. These mothers adapted to the situation and cared for the baby at home or with help from those nearby. References to God's decision, Allah's will or other destiny explanations combined with practical and economic obstacles often justified staying at home:

> I knew he wouldn't be healed if we took him to the hospital. So I left his fate to Allah and let him stay at home. . .. (Interview, mother experiencing newborn death, rural, 17)

Mothers we met in interviews described how other family members intervened in care-seeking for the ill baby. As mothers they had little say and could only accept the situation as it was. One mother explained:

> It was only me who wanted to go. No one said "go to the hospital". She [her mother] didn't tell me to go—because the baby was getting tired and not taking the breast. I thought if she would get glucose she would get better. But no one said anything, I was all alone. (Interview, mother experiencing newborn death, rural, 9)

**Manoeuvring and avoidance.** Not every mother found the advice given by the elderly or husband useful. Some mothers who had a different opinion than the husband or others providing advice found alternatives to avoid recommendations which they did not agree with.

Young women in urban areas tended to hesitate when family members recommended to use traditional medicine.

> They [the older women] tell us that during those earlier times when medications were not available, they used herbal medicine and criticise us for running to the hospital. But the current generation prefers to take their children to the health center or hospital rather than listening to their advice. (FGD; urban women)

Mothers found ways of manoeuvring care-seeking in order to provide what they found to be better care. This occurred when disagreeing with members both inside and outside the family. A woman explained how she did not trust the advice she received from the mother of her brother's wife. In order to protect her baby, she actively produced excuses to avoid adhering to recommendations that she deemed useless or even harmful.

> She [her brother's mother-in-law] was concerned about some red discoloration on the baby and asked me what happened. I replied that it's just some kind of hot weather related abnormality. Later she brought some kind of leaf that grows near the construction walls. She told me to massage the affected area, but I just put it down. She insisted that I had to apply it now, but I refused saying that I had not goven the baby a bath yet. I did that intentionally and had washed him already. I told her that I would apply it after I have showered the baby and after cleaning the leaf. I told her I'd prefer to let him sleep for now and that I would apply it later. I did that intentionally, and took him to hospital the next day... (FGD, urban women)

In interviews and FGDs, it became clear that younger women in urban areas at times hesitated when receiving advice from the elderly. They worried about the common advice to "wait and see" or even to not visit health faciltes. Some mothers dealt with this by withholding information or by seeking health care without telling anybody that the child was not well. One mother explained how she deliberately avoided asking neighbours for advice when her baby fell ill, as she would feel obliged to follow their recommendations.

> No, I didn't discuss this with my neighbours. I didn't even tell them. When you tell this to mothers, they might advise you to use traditional medicine or something similar... It's better to discuss the issue with educated people. As these people are elderly, it might be very difficult to say no or refuse to do what they ordered you to do, that's why I didn't tell them... They usually say "it's better if he takes medicine at home." Even though they didn't tell me to do it, they have advised other people to give birth at home. (Interview, mother experiencing newborn illness, rural-urban area, 13)

**Action and opposition.** At the hospital, we met mothers who alone or in collaboration with their husbands had sought care urgently when recognising that the baby was ill. Beyond negotiation, some mothers took action when they did not trust the advice they had received. In our material, cases where mothers insisted on seeking care were typically from households in urban areas, where mothers commonly had fewer children and possibly a small income. These mothers tended to dismiss the recommendations to "wait and see" and to use traditional medicine, and preferred taking the ill baby for examination by health professionals. They reiterated recommendations from TV and health care workers to immediately seek health care if the baby fell ill.

> The others [neighbours] just recommend traditional medicine rather than taking the child to the health center. Since we are aware of their sickness from what we hear from TV and have gotten education, we immediately take the children when they are sick. (FGD, urban women)

However, opposing recommendations from husbands and the elderly put women in distress. When knowing that seeking care meant high costs for the family, mothers as fathers worried about the financial impacts on the family. Health care workers reported cases where they tried to convince parents (often the father) to stay in health facilities when they wished to leave the hospital. The health care workers described cases of mothers coming to the clinics in great despair and without money and the additional burden of opposing the father's view.

> Three or four days ago a woman came with her four months old child. Her husband is a labourer and they are poor. The child had respiratory problems, the condition was severe, and we referred them to the hospital. Suddenly the mother started crying. She was distressed because she only had 10 birr [0.4$] and that was not even her money. Her husband gave her that money for another purpose and he had told her not to take the child to the hospital. He had told her that he would give the child traditional medicine. (Interview, health care worker, urban, 23)

These different responses indicate how mothers navigated the different care-seeking options that were available to them. Facilitating factors, such as financial resources or access to information, enabled some of the mothers to participate in decisions following recognition of illness, including active decision making outside the household.

## Discussion

### Bargaining power, agency and room of action

This study describes mothers' roles in decision making in the resource-constrained setting of Butajira. Decision making was a collective process, where the mother often was the first to recognise that the baby was ill and shared that information with the father and other family members (first decision point). Actual care-seeking decisions about whether (second decision point), how and where to seek care (third decision point) were however expected to be made by the father. Beyond the household, relatives, neighbours and the elderly acted as direct and indirect advisors. Mothers' decision making power was embedded in existing power dynamics at household level, and mothers showed different degrees of agency in these processes: Where some adapted to the current situation and lack of care seeking options, others took intentional actions, and even challenged what was expected of mothers. We identified acceptance and adaptation; negotiation and avoidance; and action and opposition as different care-seeking strategies mothers took on. The mothers' strategies depended on their bargaining power and room of action, which were bounded by gendered and socio-economic structures.

The findings of this study contribute to the literature on how social relationships affect recognition and responses to child illness. Colvin et al. outlined a conceptual model of household decision making that is of relevance [17]. They described four modes of household recognition and responses in care-seeking for ill children and how these are influenced by individual and contextual factors: (1) caregiver recognition and response, (2) seeking advice and negotiating access in the family, (3) making use of community-based treatment options, and (4) accessing formal medical services at health facilities. The four modes outlined in the model [17] were not available to all mothers in Butajira, where mothers' responses in care-seeking depended on

their agency, social negotiation, financial constraints and gendered norms and structures. Whereas health care policies focus on urgency in getting medical care, this was not even a considered response for some of the mothers in Butajira. The mother and her baby's opportunities were connected to her agency and bargaining power within and outside the household. Mothers were expected to be at home following birth and were rarely thought of as main decision makers on care-seeking for ill babies. Without access to financial means herself, decisions to seek formal care and pay for costly health care services hinged on the family breadwinner. In this way, mothers' care-seeking was found to be bound by gendered expectations on who makes decisions and lack of financial bargaining power. Financial resources is known to be important in Ethiopia as elsewhere, when the economic burden of drugs, diagnostics and in-patient services largely fall on families [24,26]. Social norms and the vulnerability of the newborn also influence decision making [27]. With uncertain survival chances investments in care for young babies was seen as riskier than for older children and adults. The reluctance to spending scarce resources on ill newborns alongside mothers' limited bargaining power hamper care-seeking for newborns in need of care.

This study also illustrates cases of mothers showing agency and strategic use of their bargaining power. Younger and urban women with access to information about illness aimed to influence decision making, care-seeking or took action themselves. Though worrying about high costs and consequences for the family, these mothers strategically navigated based on the various options that were available. These cases mirror other studies describing "subtle and overt ways to participate in decision-making" [14,15]. In aiming to provide "good care" for their babies, some mothers in Butajira circumvented local expectations and existing power structures and dynamics. In the interface between social norms and new knowledge, mothers balanced respect for the elderly's advice against the baby's needs and urgency for health care. These findings highlight how health information provided by TV campaigns and health extension workers was recognised, trusted and perceived as relevant by young, semi-urban women and influenced their care-seeking strategies.

## Mothers as policy targets: Opportunities and pitfalls

Our results illustrate some concerns regarding the global health agenda's emphasis on the interconnection between mothers and children in health care delivery. Despite attention to the important roles of communities and community health programs [6,8], there has been less focus on the implementation of these policies. We argue that current policies depend heavily on linear understandings of health care-seeking and decision making. In simple terms, policy documents expect a mother to immediately seek care at health facilities after identifying that the baby is ill.

Our findings align with previous research highlighting that decision making on whether and where to seek care is not a straightforward process [10–12,14,17,25,27]. While this recognition complicates how we think about health care delivery, it is vital to recognise that context matters in implementation processes. Newborn health policies provide accurate descriptions of the biomedical problems [5–7], but challenges in care-seeking go well beyond the health care sector. While efforts to make mothers responsible for making good choices for their babies are well-intended, it remains unrealistic to expect mothers to do so when they lack bargaining power or financial resources. The Ethiopian strategies to achieve universal health coverage, with emphasis on financial risk protection [32], is one step in removing financial barriers in care-seeking and decision making.

Despite recent female leadership and initiatives to empower women in Ethiopia, there is a long way to go to improve gender equality. Women's roles and opportunities to make

decisions are influenced by gendered divisions of labour, economic opportunities and risk of interpersonal violence [21,22,33]. Our study highlights that on the ground, women's room of action remains constrained. It is crucial to provide information about healthy practices to mothers, fathers and other community members taking part in decision making, while concurrently addressing underlying drivers, social processes and power structures that prevent mothers from taking active roles in everyday life and care-seeking.

## Methodological concerns

Some important methodological considerations should be noted. The study was not initially designed to explore the role of mothers in decision making; however, the theme emerged following analysis and in-depth discussions on decision making and intra-household resource allocation [26,27]. Most interviews with primary care-takers were done with mothers or in conversation with fathers and mothers (Table 2). Only one interview was done with a father alone. Further exploration of other family members' roles, including the parenting roles of fathers [34], is highly relevant for our study.

The data was analysed using thematic analysis. While it is a limitation of the study that one author led the initial coding, a different author supported in evaluation and re-evaluation of the codes. All authors were were involved in the analysis, interpretation and development of sub-themes and themes (phases III-VI), which is likely to have enhanced the trustworthiness and credibility of the study.

As a context-specific qualitative study, the findings may not be generalizable. We still argue that the findings are relevant for planners and researchers evaluating policy implementation in Ethiopia in particular, but also in similar resource-constrained contexts. Our questioning of mothers' roles and strategies in decision making may be of interest for other health policies. Further studies on how policies unfold at the local level and how they and their priorities are understood at household levels and by health care workers is highly relevant. While the latter is beyond the scope of this paper, research on how health care workers adapt and implement policies remains important [35].

## Conclusion

Health policies targeting health care workers and primary care-takers aim to increase use of health care services for newborns in Ethiopia, where health care coverage remains low. Based on our examination of mothers' roles in decision making and their strategies in care-seeking, we question the special focus on mothers in many policies and donor programs. While a mother may navigate potential care-seeking options, her bargaining power, agency and cultural and structural constraints shape her roles and strategies in decision making. We argue that the reliance on mothers as sole decision makers is simplistic and that the family as a decision making unit must be put at the center. For policies to meet their aims, the gendered social processes, financial power structures and systems in which policies are implemented must be addressed.

## Supporting information

**S1 File. Interview guides and focus group guides.**
(PDF)

## Acknowledgments

We are grateful to the participants who made this study possible. We thank Muluken Gizaw support in data collection and initial analysis, and the Butajira Rural Health Program for their assistance and facilitation of data collection.

## Author Contributions

**Conceptualization:** Kristine Husøy Onarheim, Karen Marie Moland, Mitike Molla, Ingrid Miljeteig.

**Formal analysis:** Kristine Husøy Onarheim, Karen Marie Moland, Mitike Molla, Ingrid Miljeteig.

**Investigation:** Kristine Husøy Onarheim, Ingrid Miljeteig.

**Methodology:** Kristine Husøy Onarheim, Karen Marie Moland, Mitike Molla, Ingrid Miljeteig.

**Supervision:** Karen Marie Moland, Mitike Molla, Ingrid Miljeteig.

**Writing – original draft:** Kristine Husøy Onarheim.

**Writing – review & editing:** Kristine Husøy Onarheim, Karen Marie Moland, Mitike Molla, Ingrid Miljeteig.

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
