## [Decision Letter · Decision Letter 0]

18 Nov 2019

PONE-D-19-20757

“I wanted to go, but they said wait”: Mothers’ bargaining power and decision strategies when newborns fall ill in Ethiopia

PLOS ONE

Dear Dr. Onarheim,

Thank you for submitting your manuscript to PLOS ONE. After careful consideration, we feel that it has merit but does not fully meet PLOS ONE’s publication criteria as it currently stands. Therefore, we invite you to submit a revised version of the manuscript that addresses the points raised during the review process.

We would appreciate receiving your revised manuscript by Jan 02 2020 11:59PM. To enhance the reproducibility of your results, we recommend that if applicable you deposit your laboratory protocols in protocols.io, where a protocol can be assigned its own identifier (DOI) such that it can be cited independently in the future. For instructions see: http://journals.plos.org/plosone/s/submission-guidelines#loc-laboratory-protocols

We look forward to receiving your revised manuscript.

Kind regards,

Jennifer Yourkavitch

Academic Editor

PLOS ONE

Journal Requirements:

2. Please include a copy of the interview guide used in the study, in both the original language and English, as Supporting Information, or include a citation if it has been published previously.

4. Please include a title for Table 1.

Additional Editor Comments:

Please respond to the reviewers' comments and make appropriate revisions. Please be sure to thoroughly review and edit the manuscript for English grammatical accuracy and correct usage.

Reviewers' comments:

Reviewer's Responses to Questions

**Comments to the Author**

1. Is the manuscript technically sound, and do the data support the conclusions?

Reviewer #1: Partly

Reviewer #2: Yes

2. Has the statistical analysis been performed appropriately and rigorously? 

Reviewer #1: I Don't Know

Reviewer #2: N/A

3. Have the authors made all data underlying the findings in their manuscript fully available?

Reviewer #1: No

Reviewer #2: No

4. Is the manuscript presented in an intelligible fashion and written in standard English?

Reviewer #1: No

Reviewer #2: No

5. Review Comments to the Author

Reviewer #1: This paper employs a qualitative assessment of women’s roles in making decisions about and seeking care for ill newborns. The paper drew from interviews and focus group discussions from mothers and community members and conclude that women take various approaches in care seeking. The topic is of importance to policy and program stakeholders and draws attention the need to consider women’s empowerment-related barriers to seeking care for their newborn in the design of recommendations and programs.

Although there are a multitude of factors that are at play in empowerment, the paper focuses mostly on the different types and mechanisms of decision-making thereby downplaying the role of other critical factors, for example, poverty and financial constraints, especially in the results section. Of additional concern is the lack of clarity in the in the methods section. It should be noted up front, with a forthright description of the circumstances under which the interviews and focus groups were conducted, with more than just the citation of previously published articles based on the same study. Of lesser but nonetheless important note, throughout the paper, the writing can be made more succinct, precise, and should undergo a rigorous content and copy-editing.

Major points:

A. The literature review is not entirely convincing of a gap in the literature or for the need of the paper.

B. Line 158 the authors say 41 in depth interviews, but it looks like there are 21 based on Table 1. Also, what is the breakdown of number of people and type of person (health workers, community members, care takers) in each FGD? Were the focus group participants the same as the in-depth interview participants? If the groups were homogenous, were there different questions for different type of people (health workers, community members, care takers)? Is there an appendix of questions asked? Over what period of time were these FGDs/interviews conducted? Some quotes were described by urban/rural locality- was there disaggregated analysis done? If so, that needs to be described in methods. Even if the study design is described elsewhere, this section needs more information and to be clarified. Perhaps of the utmost concern is that data collection was conducted for another study, the main study objectives or potential implications of this were barely mentioned.

C. The results section was not clearly organized, were not always well supported by quotes, and the interpretations presented at times misrepresented the issue. For example:

a. In section on identifying problem: there is no quote at all. The subject in this paragraph seems to switch from women to a specific woman. It should more explicitly state whether these were themes or based on single woman’s experiences.

b. In section on depending on decisions: the interpretations of the quotes presented and/or the entirety of the discussions diminish the power dynamic. What comes across in the quotes is that women do not have the authority in the relationship to decide to seek care. Yet the author uses words like “interdependence” (line 233) and “the husbands role made him responsible” (line 235) and “decisions where and how to seek care were not necessarily theirs to make” really undermine and downplay the disempowerment issues that are elucidated here. It’s a power dynamic that, based on the quotes presented, should be highlighted.

c. Line 258- add in older relatives or neighbors “thought they” knew whether to seek… It appears they didn’t actually know if the baby was actually sick and were advising not to go because they misunderstood medicine.

d. Quote starting on line 323 isn’t relative to taking care of a sick baby, it’s about not getting to the facility in time for delivery, and what sounds like a quick labor. I wouldn’t include the quote here, it’s not relevant to the focus of the paper.

e. Maneuvering and Action and opposition sections: it’s unclear in these two sections what applies to which place of residence. Importantly, the mention in these two sections begs the question about urban/rural differences in other sections.

D. The discussion could be more succinct and organized, with additional emphasis on the confluence of factors that relate to women’s status, empowerment, and ability to seek care, including poverty, education, urban/rural location, and access to care. This is briefly mentioned lines 497-505 but that seems to be the crux of the study.

Minor points:

1. Introduction: Although I am not completely familiar with the literature on the topic, I would assume this topic-women’s empowerment as it relates to care-seeking has been well-studied. The article mentions three distinct studies but spends only one sentence on tying them together. I question if this is a comprehensive review, and if more summary on the topic including other relevant literature can be added.

2. Suggest moving the Study Context section from Methods to Intro

3. What does treating illness through community mobilization mean ln 135-136?I see this as deployment of CHWs, not necessarily community members that are discussed in this paper, so it seems like this section may be well suited in the recommendation as a way to combat the way the culture interferes with seeking care.

4. In Table 1, the stars don’t really align with the words that they’re placed next to. It’s confusing to read like this, maybe they should be placed next to the respective column heading, and then the total number of primary care takers can be a line at the bottom of the table.

5. There are many instances in which the choice of words was questionable, which may either be a language/translation issue or stylistic, for example, using the word stakeholder to reflect care takers or community members (stakeholder more commonly refers to individuals or agencies invested in a program or policy). In other instances, there were issues with subject-predicate agreement, for example, line 274 (decisions to seek care aren’t costly, but medical care can be). Regardless, the paper should undergo more rigorous content and copy-editing.

6. FGD spelled out in some places, acronym used in others

7. In the discussion, lines 425-459 is one big paragraph that should be broken out by key messages the authors are trying to convey that highlight the main findings of the study.

Reviewer #2: The manuscripts makes an important contribution to the literature on the participation of mothers in the care-seeking process of ill children. However there are several aspects of the manuscript that require improvement, revision and clarification.

Introduction

-Overall the writing quality can be improved - there is need for further editorial and grammatical corrections.

-There is need for consistency in terminology or definitions of unclear terminology. For example, line 74 states that there is emphasis on the mother to 'improve newborn health', yet the prior discussion was on care-seeking, with the aim to improve care-seeking behavior.

-Line 77-78- define what is meant by health behavior

-Line 111-114, are there any citations to support this statement

-Line 126-127, 71% seems high given the context of the rest of the discussion, is it possible to provide additional context based on this reference

Study design

-who were the informants and how were they identified

-what was the decision-making process of following some of the primary care takers home and not others, how was the determination made on who would be followed vs not

-how were FGD participants selected

-what was the composition of FGD participants, at minimum- male vs. female breakdown

-provide more description on the thematic analysis, how was it conducted ?

-was there a documentation process of the discussions that following interviews

-was a codebook developed? what was the process of developing the codebook?

-what was the coding process? were there multiple coders? if so, what was the process of determining intercoder reliability ?

-need more clarify on line 184- 'findings influenced by earlier interpretations'

-there is need for more description on the consent and ethical review process, what type of consent did participants decide

Results

-line 244 - who tended to be the breadwinner- outside of the husband

-line 249 - most often men decide whether to go to the hospital?

-line 297-298- any more explanation of how the strategies were identifies- was there any differences strategies based on respondent type

General-

-was there is an effort to distinguish between women that experienced a newborn death vs. those that did not- it is not clear if there are differences in their experiences of these women in terms of care-seeking behaviors?

- any differences in views by FGD participants ?

6. PLOS authors have the option to publish the peer review history of their article (what does this mean?). If published, this will include your full peer review and any attached files.

Reviewer #1: No

Reviewer #2: Yes: Lwendo Moonzwe Davis

---

## [Author Response · Author response to Decision Letter 0]

10 Jan 2020

Please see attached cover and rebuttal letter which responds to the specific reviewer and editor comments.

---

## [Decision Letter · Decision Letter 1]

9 Mar 2020

PONE-D-19-20757R1

“I wanted to go, but they said wait”: Mothers’ bargaining power and strategies in care-seeking for ill newborns in Ethiopia

PLOS ONE

Dear Dr. Onarheim,

Thank you for submitting your manuscript to PLOS ONE. After careful consideration, we feel that it has merit but does not fully meet PLOS ONE’s publication criteria as it currently stands. Therefore, we invite you to submit a revised version of the manuscript that addresses the points raised during the review process.

We would appreciate receiving your revised manuscript by Apr 23 2020 11:59PM. To enhance the reproducibility of your results, we recommend that if applicable you deposit your laboratory protocols in protocols.io, where a protocol can be assigned its own identifier (DOI) such that it can be cited independently in the future. For instructions see: http://journals.plos.org/plosone/s/submission-guidelines#loc-laboratory-protocols

We look forward to receiving your revised manuscript.

Kind regards,

Jennifer Yourkavitch

Academic Editor

PLOS ONE

Additional Editor Comments (if provided):

Thank you for this revision, which addressed many of the reviewers' comments. There are still serious concerns about the analysis, which does not conform to qualitative research standards. At a minimum, please invite a second person to code and analyze the data so that there is some demonstrable validity to the results and conclusions. Please also create and use a code book. And then describe fully the methods you used to re-analyze the data. Describe the themes that emerged from the analysis. Please address and respond to all of the reviewer's comments.

Qualitative data can be made publicly available without identifying information. You can provide tables or matrices with raw information, organized by themes.

Reviewers' comments:

Reviewer's Responses to Questions

**Comments to the Author**

1. If the authors have adequately addressed your comments raised in a previous round of review and you feel that this manuscript is now acceptable for publication, you may indicate that here to bypass the “Comments to the Author” section, enter your conflict of interest statement in the “Confidential to Editor” section, and submit your "Accept" recommendation.

Reviewer #2: (No Response)

2. Is the manuscript technically sound, and do the data support the conclusions?

Reviewer #2: Partly

3. Has the statistical analysis been performed appropriately and rigorously? 

Reviewer #2: N/A

4. Have the authors made all data underlying the findings in their manuscript fully available?

Reviewer #2: No

5. Is the manuscript presented in an intelligible fashion and written in standard English?

Reviewer #2: No

6. Review Comments to the Author

Reviewer #2: The authors have made significant improvements to the manuscript, there are noteworthy improvements to the introduction, discussion and conclusion. However, the primary concern with the manuscript upon further review is the lack of methodological and analytical rigor, which is not of the standard required for publication. The authors note that a codebook was not developed and the coding done by one author with discussion by team. There is limited discussion on the process used for coding and analysis and thus it is not possible to assess the validity of the process.

Specifically the reliability and validity of the research findings are called into question without the details of the analytical process. For example, the authors discuss conducting thematic analysis however it is not clear which themes emerged, are the only themes that emerged those that are discussed in the paper? Were there other themes that were not included, if so what was the process for deciding what would be included? Since coding was conducted by a single author, would the same themes be evident to others? Was there an attempt to create objectivity in the analysis? Where any of the analytical tools that are available in Nvivo utilized for the analysis? Were interviews recorded and transcribed? Or were transcripts based on notes? What was the quality of these transcripts? Where there any data quality related challenges in the transcription and translation process? If so how were these mitigated?

Although the authors have not adequately described there analytical process, they have described noteworthy findings and conclusions. Yet without this foundational aspect, the one cannot sufficiently assess the validity of these findings and conclusions.

7. PLOS authors have the option to publish the peer review history of their article (what does this mean?). If published, this will include your full peer review and any attached files.

Reviewer #2: No

---

## [Author Response · Author response to Decision Letter 1]

30 Mar 2020

Please see cover letter which includes detailed responses to the reviewer and editor comments.

---

## [Editor Report · Decision Letter 2]

7 Apr 2020

PONE-D-19-20757R2

‘I wanted to go, but they said wait’: Mothers’ bargaining power and strategies in care-seeking for ill newborns in Ethiopia

PLOS ONE

Dear Dr. Onarheim,

Thank you for submitting your manuscript to PLOS ONE. After careful consideration, we feel that it has merit but does not fully meet PLOS ONE’s publication criteria as it currently stands. Therefore, we invite you to submit a revised version of the manuscript that addresses the points raised during the review process.

We would appreciate receiving your revised manuscript by May 22 2020 11:59PM. To enhance the reproducibility of your results, we recommend that if applicable you deposit your laboratory protocols in protocols.io, where a protocol can be assigned its own identifier (DOI) such that it can be cited independently in the future. For instructions see: http://journals.plos.org/plosone/s/submission-guidelines#loc-laboratory-protocols

We look forward to receiving your revised manuscript.

Kind regards,

Jennifer Yourkavitch

Academic Editor

PLOS ONE

Additional Editor Comments (if provided):

Thank you for providing more information about the analysis. Since you did not re-analyze the data with a second coder as suggested ("supported by" does not indicate two independent coders), please include a Limitations section (perhaps renaming and expanding the current "Methodological Considerations" section) that acknowledges a potential lack of objectivity in the analysis due to reliance upon a single coder. You could also state whether you believe that input from co-authors during other steps of the analysis enabled objective discussions of the results.

Please do not indicate who did what in the Methods section of the text; remove the initials of individuals. It is appropriate, however, to indicate where all authors participated in a step, and also that the coding was done primarily by one person.

---

## [Author Response · Author response to Decision Letter 2]

1 May 2020

Please see attached response letter/cover letter responding to the editor's comments and the data availability statement.

---

## [Editor Report · Decision Letter 3]

11 May 2020

‘I wanted to go, but they said wait’: Mothers’ bargaining power and strategies in care-seeking for ill newborns in Ethiopia

PONE-D-19-20757R3

Dear Dr. Onarheim,

We are pleased to inform you that your manuscript has been judged scientifically suitable for publication and will be formally accepted for publication once it complies with all outstanding technical requirements.

With kind regards,

Jennifer Yourkavitch

Academic Editor

PLOS ONE
---

## [Editor Report · Acceptance letter]

28 May 2020

PONE-D-19-20757R3 

‘I wanted to go, but they said wait’: Mothers’ bargaining power and strategies in care-seeking for ill newborns in Ethiopia 

Dear Dr. Onarheim:

I am pleased to inform you that your manuscript has been deemed suitable for publication in PLOS ONE. Congratulations! Your manuscript is now with our production department. 

With kind regards,

on behalf of

Dr. Jennifer Yourkavitch 

Academic Editor

PLOS ONE